# Fuzzy Quality Certification of Wheat

Cristian Silviu Simionescu [1], Ciprian Petrisor Plenovici [1], Constanta Laura Augustin [1,*], Maria Magdalena Turek Rahoveanu [1], Adrian Turek Rahoveanu [2] and Gheorghe Adrian Zugravu [1]

[1] Faculty of Engineering and Agronomy in Braila, "Dunarea de Jos" University of Galati, 800008 Galati, Romania
[2] Faculty of Management and Rural Development, University of Agronomic Sciences and Veterinary Medicine Bucharest, 011464 Bucharest, Romania
* Correspondence: laura.zugravu@ugal.ro; Tel.: +40-722-676-202

**Abstract:** This paper presents a fuzzy quality certification of wheat. This analysis is based on the fuzzy analysis model of wheat. We developed a Matlab application with the help of which we modeled the perceptions in relation to the main quality physical and chemical characteristics of wheat obtaining a quality index of wheat lots. The algorithm presented in this article allows for obtaining and using the global quality index, generating applicability not only to the commercial sphere as a quality reference and price setting, but also a measure of appreciation of processing opportunities. Indices of fuzzy quality associated with wheat lots using a fuzzy model offer the opportunity to develop local markets through quality certification.

**Keywords:** wheat quality; fuzzy quality certification model

## 1. Introduction

The main objective of this paper is to develop an informational model for the standardization of wheat quality in international contracts and obtain a global quality index of the lots analyzed. Decisions about the quality of wheat lots involve imprecise and vague information [1,2]. The development of the information model is based on the assessment by experts of the results obtained from the analysis of 25 batches of wheat, through the reference methods, and the transposition of these results into a mathematical modeling system, called a fuzzy system. By using the fuzzy system, the quality of the analyzed lots will be translated into a global quality index related to each lot. Fuzzy systems are oriented towards managing uncertain or imprecise information [3,4]. The fuzzy system is used in fields where the input variables do not have fixed values or their value and importance may vary [5,6]. The fuzzy concept was first introduced by Zadeh in 1965, to alleviate uncertainties and fuzziness problems. The technique relies on human subjectivity in decision-making due to linguistic variables that allow precise modeling of imprecise entrances [7,8] and has been applied to many engineering problems [9,10]. This concept offers a number of advantages for users, such as reducing the costs of production, transport, storage, and recovery as well as improving the costs of the companies using it [11,12]. Fuzzy is used to represent process uncertainty and simulation of final product quality determination [13–15]. Fuzzy logic is a broad field of study and various tools have been developed in recent years. Food quality is a fuzzy category that could be evaluated using fuzzy logic [16–18]. Its implementation in food quality control for the food industry has been highlighted by several authors who have focused on different applications designed specifically for this field [19,20]. This is especially true when considering the reasoning process, expressed in linguistic terms, of operators and experts [21,22]. However, applications are still limited and few reviews are available on the subject [23–25]. By using fuzzy logic, obtaining the global wheat quality index can be determined by going through the following stages in the development of the informational model:



- Formation of the knowledge base;
- Fuzzy inference;
- Fuzzification;
- Defuzzification.

The adopted approach regarding the evaluation of wheat quality is based on the expertise of 20 specialists in the field, with competences in the determination and evaluation of wheat quality as well as in the field of grain trading. The allocation of qualifications by experts regarding the quality of the lots analyzed, as well as the establishment of the weight of the importance of the analyzed parameters, was carried out based on international and European standards, but also on the quality specifications used in commercial contracts, together with the professional experience of each specialist. Mapping quality attributes into a fuzzy domain as multidimensional fuzzy sets results in a quality index associated with the entire lot [26–28]. In quality control, specialists may face uncertain and unclear concepts. By using the fuzzy concept and developing the information model, the quality of the wheat is qualified according to the strictness of the decision factors and the values of the obtained quality parameters. Natural activities and human thought form the basis for fuzzy logic that presents itself based on various application perspectives [29,30]. At present, food safety incidents have occurred frequently in China and customer trust has declined rapidly;therefore, food quality and safety issues have attracted more and more social attention [31,32]. Considering the concern about ensuring food quality and improving consumer confidence, many companies have developed a traceability system based on the fuzzy concept to visualize the supply chain and avoid food safety incidents [33–35]. Fuzzification processes are implemented in the fields of health care, education, career selection, real estate, and financial markets [36–38].The parameters used for systems analysis are the input factors, the type of membership function used for fuzzification, defuzzification of the generated fuzzy sets [39–41]. Based on the results generated by this system, information is generated that derives recommendations for the selection and optimization of processes. Agriculture is considered as a system that provides products of value and indeterminate returns. It is essential to choose an appropriate technique to maximize yield and minimize losses in the trade chain.

## 2. Methodology

In this paper, we proposed a model, which can not only perform a quality assessment at all control points, but also assess the quality of wheat that is the subject of an international contract. The quantities that are the subject of commercial transactions usually come from several farmers and are stored before delivery in several storage areas (platforms, warehouses, and silo cells). The use of lot mapping based on the resulting quality index is an advantage for traders and storekeepers, generating a clear and objective overview.

Wheat quality is a complex and widely used term to describe the ability and general potential of wheat to be used in a wide variety of finished products by milling and obtaining quality flours for the production of bread and bakery productsand pastry, semolina, as well as the use in various processes in the extractive, fermentation industries, or in the animal husbandry industry.

The main determinants of wheat quality are endosperm texture (grain hardness), protein content and gluten concentration.Endosperm texture in wheat is the single most important and defining quality characteristic, as it facilitates wheat classification and influences milling, baking, and end-use quality [42].

For millers, wheat quality is considered to be the ability of a wheat variety to produce high quantities and qualities of flour or semolina during the extraction process.In this process, the level of contamination of flour or semolina with bran fractions is also important and is related in most cases to undesirable characteristics for the end-use quality of the productas well as grain hardness [43].

Millers prefer large, uniform, whole, unpolished, and full grains.These physical characteristics, along with chemical and rheological properties, are objectives for wheat growers to increase yield quality and production [44].

Sampling represents the operation that consists of taking and constituting a sample in order to determine the quality by analyzing the monitored parameters.The sample must be as representative as possible for the sampled lot.Sampling is carried out both for grain in motion and for batches of stationary grain or in packaged units (bags).

In the case of bulk, stationary grains, elementary samples are taken with the manual cylindrical probe from different points. From the point of view of the regulations regarding the approach to the sampling process, ISO launched the specific standard for sampling, a document in circulation under the name ISO 24333:2010—Cereals and cereal products. Sampling has beentaken up by most National Standardization Bodies. Depending on the size of the lot, the standard provides the mass of the elementary sample, the minimum number of elementary samples, the minimum mass of the laboratory sample, and limits the maximum size of the lot to 1500 tons [45].

Depending on the analyzes requested, the mass of the laboratory sample may be higher, taking into account that for the determination of aflatoxin and ochratoxin a quantity of 10 kg is needed. The determination of other contaminants such as heavy metals, pesticides, DON (deoxynivalenol), or dioxins can also be identified in samples of at least one kilogram, and for fumonisins and zearalenone a quantity of 3 kg is required. The mass of the laboratory sample is determined according to the required determinations of contaminants to which is added the minimum mass provided by the standard.

The evaluation of the organoleptic and sanitary characteristics is established already in the pre-harvest phase, to avoid possible contamination of the installations or storage spaces. Organoleptic characteristics consist of appearance, color, smell, and taste.Determining the organoleptic characteristics of wheat grains are the examination through the sensory organs of qualified personnel.

International standard ISO 7971—Common wheat specifies and establishes the organoleptic conditions of the wheat grains and stipulates that the grain mass subject to evaluation must be free-flowing, without foreign smell and taste that would indicate a change in the product mass (moldiness or burning), with a normal appearance and a characteristic color [46].

The appearance is determined by visual analysis of the laboratory sample spread in a uniform layer on a white tray to allow the observation of deviations from the specific appearance. The evaluation is based on the shape of the seeds, if the grains are well developed, mature, and healthy or if they are shriveled, burnt, sprouted, altered, attacked by insects or diseases, etc. The uniformity of the grains and the appearance of the skin are monitored.

Maturity means reaching the complete and stable physiological stage. The normal appearance of the bean is considered when the covering of the bean has not undergone changes due to adverse weather conditions, improper storage conditions or attack by insects or other pests. Determining the color consists of assessing it in natural light, observing any changes compared to the characteristic color of the product. The change in the color of the berries can be influenced by excessive humidity, heat, spoilage, mold, drying or improper storage, and contact with chemical substances. Odor determination can be performedboth for whole grains and ground seeds. Determining the smell of whole grains is performedby heating and rubbing in the palms of about 100 g of seeds and inhaling immediately. Another method consists of putting 10–20g of wheat in a glass of warm water with a temperature of 60 °C, which is covered and left to rest for 2–3 min, after which the resulting vapors are inhaled, and then the water is removed from the glass and examine the smell of the remaining grains. Following the procedure used, it is assessed if the smell of the sample is characteristic, in accordance with the specification in the product standard or if it presents certain changes that may come from an inadequate storage without ventilation, from a heating of the product mass or from mold colonies. It is also possible to identify

the smell of putrefaction, of decay, of rancidity, of fermentation, honey (in case of mite infestation), musty, of foreign substances such as phosphine, fuels or sulfur, the smell of strongly aromatic plants if they are present seeds of these plants in the mass of the product or other foreign smells. Appreciating the taste is performed by chewing a few grains of wheat, preferably ground, after removing impurities and spoiled grains. The analysis aims at the specificity of the taste and if it corresponds to the specifications of the quality standards or, on the contrary, is it bitter, sour, hot, or rancid. This determination is not performed on altered, moldy grains, on those that show traces of entomological attack, or on those treated for the purpose of seeding or to combat pests. It is important to know the origin of the batches in order to avoid ingesting chemicals from fertilizers or other agrochemical products and also to identify weed seeds that may contain toxic alkaloids (e.g., ricin) [47].

*Experts' Assessment of the Quality of Wheat Lots*

Using fuzzy logic to describe abstract concepts and design decision-making systems much closer to the way a human does is an interesting and useful area to explore. To effectively implement these types of systems, expert knowledge of the domain in which the application is being used is required [48–50]. For each linguistic term that a linguistic variable implies, a fuzzy set described by a relevance function will be created. The semantic properties of the (linguistic) concept are described by the outline of the respective fuzzy set [51,52]. Therefore, the closer the behavior of the phenomenon under study is to the curve of the relevance function, the more accurate or performing the fuzzy model is in representing the real world [53–55]. In the database formation stage, the physic-chemical quality indicators obtained from laboratory determinations by analyzing the samples of the 25 batches of wheat were based on the evaluations of 20 experts using a scale with 5 linguistic terms, associated with the qualifiers:

- N = unsatisfactory;
- S = satisfactory;
- M = medium;
- B = good;
- FB = very good.

The terms N, S, M, B, and FB represent the linguistic variables to which the values of the analyzed parameters are associated. Each expert evaluated the 25 batches of wheat and assigned a value of 1 on the N-S-M-B-FB qualification scale to each analyzed parameter once, the assessment being expressed in accordance with the international standards and specifications used in international trade, as well as with their own expertise. Based on relevance, the system sets each value in the fuzzy set to a value between 0 and 1, a measure that represents the degree of relevance of the fuzzy set element.

The second stage in the development of the informational model consisted in defining a scale of three qualifications, respectively little important (PI), important (I), and very important (FI), which was made available to the experts in order to assign a qualification to each parameter that was determined in order to evaluate the quality of the wheat. Based on the standards, the international specifications used in the international wheat trade, as well as their own expertise from professional activity, the experts assigned a qualifier to the quality indicators.

By using the Matlab R2020 program and the Fuzzy Logic Designer function, the association of triplets was achieved following the evaluation of the quality of wheat lots by experts, as well as for the weights established regarding the importance of quality parameters in the evaluation of a wheat lot. Fuzzy triplets were associated with the linguistic terms used to assess the quality of the wheat batches with the help of left-right triangular membership functions, as follows:

- Unsatisfactory, (N) = [0 0 25];
- Satisfactory, (S) = [25 25 25];
- Medium, (M) = [50 25 25];

- Good, (B) = [75 25 25];
- Very good, (FB) = [100 25 0].

Linguistic terms used to determine the weights of the analysis indicators in the value of the global quality index were associated with fuzzy triplets with the help of left-right triangular membership functions as follows:

- Slightly important, (PI) = [0 0 50];
- Important, (I) = [50 50 50];
- Very important, (FI) = [100 50 0];

## 3. Results

The relative weight of the physical–chemical analyses is logged in calculation of the global wheat quality index. Additionally, it was determined using the function of the application based on the following relationship [56]:

$$Ft \; = \; F \times [PI; \; I; \; FI]/20; \tag{1}$$

The fuzzy triplets associated based on the relative weight of the physic-chemical analyzes were transposed into the matrix calculation function of the fuzzy application and subjected to modeling as follows [56,57]:

$$Qt \; = \; sum(Ft(:, 1)); \tag{2}$$

$$Ftrel \; = \; Ft/Qt; \tag{3}$$

Ftrel—represents the weight matrix of each quality indicator in the calculation of the global quality index.

The calculation of the global quality index of the analyzed wheat lots includes the fuzzification and defuzzification phase. With the help of the relative weights in the form of fuzzy triplets, the global quality index was calculated for each wheat lot, using the extended pe.mat product. The mathematical model used is [58]:

$$Clti \; = \; \Sigma \; lti \otimes Ftreli, \; where \; i \; = 1 : 30. \tag{4}$$

The mathematical model uses the extended product that was introduced in the form of the pe.mat function [58]:

$$
\begin{aligned}
&\%the \; extended \; product \\
&function \; C \; = \; pe(A, \; B) \\
&\quad C(1) = \; A(1) \times B(1); \\
&C(2) = \; A(1) \times B(2) + \; B(1) \times A(2); \\
&C(3) = \; A(1) \times B(3) + \; B(1) \times A(3);
\end{aligned}
\tag{5}
$$

To calculate the quality of the wheat batch, we used the cg.mat function, which transposes the above mathematical model into the Matlab language [59]:

$$
\begin{aligned}
&\%quality \; of \; the \; wheat \; batch \; wheat \\
&function \; C \; = \; cg(A, \; B) \\
&\quad C \; = \; [0 \; 0 \; 0]; \\
&for \; i = 1 : 30, C \; = \; C + pe(A(i, :), B(i, :)); \\
&\quad end;
\end{aligned}
\tag{6}
$$

The complexity of the algorithm is given by the matrix analysis of the 30 physic-chemical indicators of the wheat batches, indicators that are translated into fuzzy triplets. Thus, the ICG vector is obtained, with quality indices for each batch.

Correlation of the global quality index by reporting to the Grading Plan for common wheat in Romania was achieved by ordering the values of the ICG index in descending

order and the association with the assigned grade. In the grading operation, the fraction of grains attacked by black point was eliminated from the value of total impurities, their identification and highlighting had the role of making a complex assessment taking into account all the fractions provided by the standard.

After analyzing the two methods of assessing the quality of some wheat batches, it can be concluded that the minimum values of ICG obtained in the case of the 25 analyzed batches were mainly attributed to the infested batches and whose determining parameters in the grading recorded values below the limits imposed on the RO 1 degree (Figure 1). The criteria that are the basis of the grading operation are the sanitary characteristics, the content of total impurities, the hectoliter mass, and the protein content relative to the dry substance.

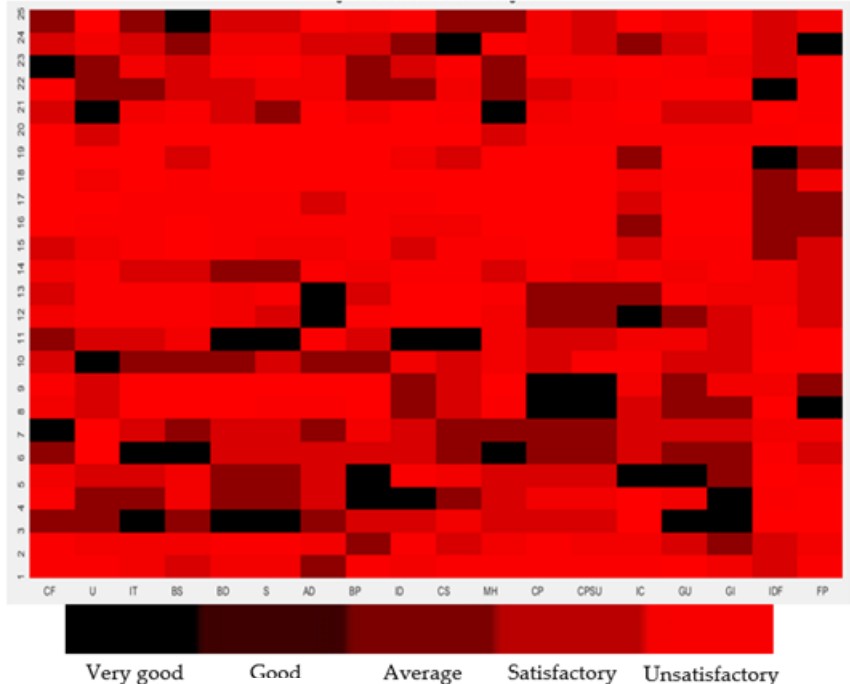

**Figure 1.** Map of the global quality indices in the analyzed commodity.

The global quality index can also be implemented in credit contracts where wheat stocks are brought as a guarantee, credit institutions having an image not only of the quantity. Researchers can use the global quality index in the study of culture technologies or breeding processes in the commercial chain; the global quality index can highlight the critical points of some batches and facilitate differentiated storage in warehouses, barges, ship holds, silos, wagons or other means of transport. From the point of view of continuing the research and development of the global quality index, the proposed informational model can be extended to other types of agricultural seeds and can be associated with an application that generates an informational model for setting the price. In the study of wheat quality, the global quality index can also be studied by resizing the number of parameters or limiting it to certain indicators of interest.

The weights of the analysis indicators in the value of the global quality index form a neutrosophic statistic, which is a generalization of the classical statistic.What distinguishes neutrosophics from other fields can be indeterminacy, neutrality, even game, unknown, contradiction, ignorance, imprecision, etc. The neutrosophic approach as a generalization of fuzzy logic is a generalization of classical probability and imprecise probability [60–62].

The substantiation of the importance of wheat from an economic and cultural point of view, as well as the quality conditions stipulated in the specific contracts used in the international wheat trade, highlighted the different perception on the evaluation of wheat quality. The concept of wheat quality in large wheat-producing countries is differently

perceived and adapted to the socio-cultural life of the analyzed regions. The appreciation and importance of quality parameters emphasizes the geographical character, the natural and economic resources available to each state, these aspects differentiate and limit the quality of certain wheat lots through national norms related to the international standards and regulations in force. In the stage proceeding the generation of the information model, applied research was carried out regarding the determination of the main quality indicators in order to identify the critical points and the aspects related to an objective, correct and representative evaluation of the analyzed wheat lots.

Methods used, as well as the results obtained, were described in relation to European and international product specifications. Throughout the work, the conditions and limitations that can classify wheat batches as unfit for human consumption, as well as aspects related to food safety through potential contamination with toxins or impurities difficult to eliminate in the conditioning, handling processes, were taken into account or transportation. All the lots analyzed can be used in baking, the resulting flour being suitable for making bread, some lots requiring mixing with quantities of a higher quality or adding additives to capitalize the flour in pastry products or other bakery specialties. The qualitative analysis of the lots reveals the need for quick intervention on the lots where live infestation was identified, by taking the necessary measures, namely their gassing or fumigation. Improving the quality of the analyzed batches can be achieved through additional conditioning, thus reducing the fractions of the category of total impurities and improving the hectoliter mass parameter. Changing the hectoliter mass involves increasing the content of extracted flour and can positively influence the parameters aimed at the rheological behavior of the dough, but also the content and quality of gluten.

In recent decades, rapid wheat quality determination methods and devices have become widely adopted for the determination of moisture content, hectoliter mass and protein in general, but also for continuous monitoring of stock quality. The analyzers are based on near-infrared (NIR) transmission technology, which can be used for the simultaneous and precise determination of several parameters, such as moisture, protein content, gluten content, fat content, but also hectoliter weight. From the halogen lamp housing on the back of the instrument, light is guided through an optical fiber into the monochromator inside the instruments. The monochromator provides monochromatic light in the spectrum from 850 nm to 1050 nm, and by means of an optical fiber, the light is guided to the collimator lens system, which is placed above the sample cup in the sample cup chamber. After the light is transmitted through the sample, the unabsorbed light reaches the detector. The detector measures the amount of light and sends the result to the digital signal processor which communicates with the computer, calculating the result. Rotating the sample cup between sample scans (called sub-samples) allows more parts of the sample to be analyzed. Sub-samples are chosen from one or two concentric circles in the sample cup, providing a more representative result from an inhomogeneous sample. Devices can be calibrated by using certified reference materials. Hyperspectral imaging (HSI) combines near-infrared (NIR) spectroscopy and digital imaging to provide information on the chemical properties of wheat grains. In order to establish and identify the wheat grains whose germination has started, studies were carried out using hyperspectral near infrared (NIR) imaging (HSI) for their detection. Experiments were conducted to determine which spectral bands have the best potential to discriminate between sound and sprouted grains. Two wavelengths were selected and combined into an index that was used to indicate the presence or absence of germination. Experiments have shown that the proposed method is effective in identifying the grains for which the germination process has started, achieving 100% accuracy for the samples used in this study. An imperfect correlation with the fall index was also observed, making it difficult to accurately determine the degree of germination, especially if the sprouts are not yet visible. These results confirm the utility of the near-infrared spectral range for detecting chemical alterations in wheat grains [63]. Protein content is one of the most important quality factors in wheat and can be determined using this technique. To solve the recognition and classification problems associated with

impurities in wheat, the researchers developed a recognition method that uses a convolution neural network. The development of this network consisted in the construction of a data set of wheat without impurities and of five impurities, with which the filtering algorithm and the enhancement algorithm were used for image pre-processing. Based on research, the testing accuracy was between 98.59% and 99.98%, respectively. Consequently, the developed network, named WheNet can be a useful tool in the recognition of impurities in wheat. In addition, this method can be used to detect impurities in other domains [64]. The determination of the total nitrogen content by combustion according to the Dumas method and the calculation of the crude protein content is based on the quantitative digestion by burning the sample at about 900 °C in excess of oxygen [65]. The sample is burned, and the organic elements are oxidized. Combustion gases ($O_2$, $CO_2$, $H_2O$, and $N_2$) and nitrogen oxides (NOx) are removed, except nitrogen and nitrogen oxides. Carbon dioxide and water are removed by passing the gases through special columns. The nitrogen content is determined by gas chromatography, and the crude protein content is calculated by multiplying the amount of nitrogen measured by the appropriate factor and expressed as a percentage.

A comparative study on the accuracy of protein determination methods, namely the Kjeldhal method, the Dumas method and the NIR technique revealed the precision error rate below 2% for the Kjeldahl method, while the precision error rate for the Dumas method varied in a range of 2–4%. The NIR method proved to be the fastest in determining protein content; however, the error rate varied between 3% and 6%. The Kjeldahl method, due to its high precision and very small ranges of variation, has made it the major method for estimating protein in food. The Dumas method for the quantitative determination of organic nitrogen was at least as accurate as the Kjeldahl method, but considerably faster. The NIR method has a relatively large standard deviation and is particularly useful for rapid analysis of protein content [64]. Both the Kjeldahl and Dumas methods for protein determination in foods are currently used, but the empirical nitrogen factors used to convert determined nitrogen content to protein content are based only on the Kjeldahl method [65].

The main objective of the research described in this article, represents an innovative method of approaching the quality of wheat and can be a landmark in the calculation of penalties and bonuses within international commercial contracts. Going through the stages described in this research has generated a global wheat quality index that can be extended for use in several commercial, governmental, or scientific segments. The research transposed in this article combined the international standards that regulate the reference methods for wheat quality determinations frequently used in international contracts, as well as national and contractual specifications in terms of quality determination. In addition to the specialized literature, which mostly includes studies on changes and behavior of wheat in different phases of culture, storage or processing, personal experience in the field of quality and the appreciation of experts on the weight of the important quality parameters and the evaluation of quality based on the results obtained, have led to the configuration of the proposed informational model. Obtaining and using the global quality index generates applicability not only to the commercial sphere as a quality reference and price setting, but also a measure of appreciation of processing opportunities. The global quality index can also be implemented in credit contracts where wheat stocks are brought as a guarantee, credit institutions having an image not only of the quantity. Researchers can use the global quality index in the study of culture technologies or breeding processes in the commercial chain, the global quality index can highlight the critical points of some batches and facilitate differentiated storage in warehouses, barges, ship holds, silos, wagons, or other means of transport. From the point of view of continuing the research and development of the global quality index, the proposed informational model can be extended to other types of agricultural seeds and can be associated with an application that generates an informational model for setting the price. In the study of wheat quality, the global quality index can also be studied by resizing the number of parameters or limiting it to certain indicators of interest.

In all lots, a higher share of seeds belonging to other plants is noted, but none of the subject lots exceed the limits of this category in terms of this sub-parameter. The values of the content of foreign bodies (chaff, dust) are reduced, which indicates a good conditioning before storage or a correct adjustment of the harvester. No toxic seeds, rye horn, or grains attacked by common wheat blight or Fusarium spp. were identified. A remarkable aspect is the lack of burnt-hot berries, which leads to the hypothesis of a moderate dryness after reception.

Sampling was carried out in accordance with the specific standard and during sampling it was possible to assess the mass of the product, not identifying agglomerations in layers or in the extracted elementary samples, the extracted wheat is free flowing. From the point of view of the organoleptic characteristics, all the lots analyzed fell within the specificity of the healthy product in terms of smell, appearance, and taste, and no alterations were identified in the mass of the product. Live infestation was observed in seven batches, the identified species being Sitophilus zeamais, Rhyzoperta domnicasi, and Cryptolestes ferrugineus. The moisture content values are between 11.27% and 13.42%; no batches with moisture above the maximum allowed limit of 14% being identified. The hectoliter mass recorded values below 77 kg/hl. Five lots (16, 17, 18, 19, and 20) have values above 77 kg/hl, reaching a maximum of 80.5 kg/hl. From the point of view of total impurity content, five samples exceed the 6% limit related to grade RO1 wheat, thus ranking the lots in terms of total impurity content in grade RO2. In the sum of total impurities, the presence of broken grains and defective grains can be noted in all analyzed batches. Sprouted grains were identified in 9 of the 25 lots analyzed, and the maximum percentage resulting from the analysis is 0.10%, so that none of the lots presents a risk of damage or a risk of failure for this reason. In all lots, a higher share of seeds belonging to other plants is noted, but none of the subject lots exceed the limits of this category in terms of this sub-parameter. The values of the content of foreign bodies (chaff or dust) are reduced in all analyzed samples, which indicate a good conditioning before storage or a correct adjustment of the harvester. No toxic seeds, rye horn, or grains attacked by common wheat blight or Fusarium spp. were identified. A remarkable aspect is the lack of burnt-hot berries, which leads to the hypothesis of a moderate dryness after reception. The use of the reference method to determine the protein content led to obtaining reliable results regarding the value of this parameter. The determination of the crude protein and then reporting to the percentage of moisture revealed variable percentages, the minimum values being 11.69 in the case of batch 19, respectively, and 11.98% in the case of batch 17. All other batches recorded values above 12%; the maximum value was13.75%. In the optimal range of 22–25% wet gluten content there are only three lots, thus characterizing the related quantities with an average wet gluten content, and in the range of 25–31% there are 22 lots of wheat with a high gluten content, thus placing all lots in higher quality classes in terms of this parameter. The values obtained after determining the gluten index parameter fall into the category of normal gluten with values between 30–80%. In the 25 lots analyzed, the drop index is above the minimum limit of 220 s. The values obtained after establishing the deformation index of this determination place 23 batches in the optimal range between 5 and 13 mm; batch 19 and 23 hadvalues below 5 mm. From the point of view of the rheological properties and the parameters determined to generate an image of these properties, only batches 4, 21, and 22 reach the optimum values, respectively, for W, G, and can be considered batches with excellent baking properties. The determination of the DON content revealed the presence of the mycotoxin in 12 analyzed lots, but the values obtained do not endanger public health, being at most 1/3 of the maximum allowed limit. In the case of 13 batches, the values obtained were below the detection limit of the device. All the results of the performed tests fell within the repeatability limit provided in the method standards used.

## 4. Conclusions

The yield and efficiency of the wheat crop is quantified both by the quantity harvested and by the quality obtained. The quality of the wheat is determined from the pre-harvest

phase by determining the moisture content to identify maturity and the optimal harvesting period, as well as by identifying possible microbial contamination or other aspects related to the physical structure of the grain. Post-harvest, quality determination is carried out in several sequences prior to processing. The primary evaluation is carried out immediately after harvesting and it is important to know the physico-chemical aspects of the grains to be able to intervene and subject the quantities received or stored to immediate drying and conditioning operations. In establishing the quality of some wheat lots, an important role is played by sampling, which must generate a sample representative of the whole lot or informative, depending on the parameters being pursued. Sampling rules are established by international, national, or trade association standards, all with the major objective of obtaining a representative sample to provide a clear overview of quality. Quality assessment is initially carried out by evaluating the organoleptic and sanitary characteristics that must correspond to the healthy product, present a free-flowing appearance of the seed mass, without agglomerations or modified color, smell, and specific taste. Humidity plays an important role, being the first determination that is made after the organoleptic analysis and determination of infestation, its value being taken as a reference in the calculation of subsequent determinations. From the point of view of physical characteristics, the hectoliter weight is a useful indicator in the milling and baking industry, and at the same time a low value of this parameter can classify the wheat in lower quality classes or can be considered fodder. Studies have shown that from a quantity of wheat with a high hectoliter mass, the amount of flour extracted is greater compared to the amount of flour obtained from wheat with a low hectoliter mass. The chemical analyzes consist of determining the protein content, the quantity and quality of gluten, but also the behavior of the flour during kneading by determining the falling index and alveographic properties. The frequent use of irrigation water from various sources and plant protection products in an uncontrolled and sometimes irrational way, the determination of the content of heavy metals and pesticides has become in recent years an important criterion for determining the quality of wheat. Non-compliance with culture technologies and climatic conditions favors the development of some species of fungi such as Fusarium spp. which, due to their toxic nature, produce secondary metabolites generically known as mycotoxins. Vomitoxin or deoxynivalenol is considered the wheat-associated mycotoxin, with studies showing little presence of aflatoxins, ochratoxin or zearalenone in wheat. The methods for determining quality indicators are varied and are regulated by international standards and regulations, these having a mandatory character in the case of heavy metals, mycotoxins and pesticides. By contributing the experts in evaluating the quality of the wheat lots based on the results of analyzes made available and by ranking the importance of parameters within a global appreciation, the database required for the mathematical modeling system was constituted.

The use of fuzzy logic in the configuration of the informational model was achieved by using the Matlab 2020 program and the fuzzy logic designer function. The Matlab program with fuzzy functions is frequently used and there are numerous scientific articles that are based on statistical data processed by this method. The specialized literature emphasizes the potential of fuzzy applications used to render the uncertainty of the process and simulate determining the quality of a final product. The principle of fuzzy logic is based on the transposition of clear results into an unclear fuzzy system, by associating triplets with value from 0 to 1 in the fuzzy stage and subsequently subjected to defuzzification, resulting in an associated value in the form of an index. The map of quality attributes by fuzzy technique as multidimensional fuzzy sets and later defuzzification resulted in obtaining a global quality index associated with the whole lot (IGC).

The limits of these parameters with an impact on food safety are established both for unprocessed wheat and for products obtained from it and intended for human and animal consumption.

The evolution of technology has allowed the development of methods for rapid determination of wheat quality in order to automate and obtain quick results, but in case of litigation the reference methods provide the most accurate results for settlement and

arbitration. The expansion in time of quality determinations, the interest of standardization organizations regarding the methods and limits of wheat quality parameters, as well as innovation, position wheat in an area of major interest, generating competitiveness in the sphere of quality and its determination as a future concern.

**Author Contributions:** Conceptualization, C.S.S., C.P.P. and M.M.T.R. investigation, C.P.P. and A.T.R.; methodology, G.A.Z. and C.L.A.; software, G.A.Z. and A.T.R.; validation, C.P.P. and M.M.T.R.; writing—original draft, C.L.A. and C.P.P.; writing—review and editing, C.S.S. and G.A.Z.; supervision, G.A.Z. All authors have read and agreed to the published version of the manuscript.

**Funding:** This research received no external funding.

**Institutional Review Board Statement:** Not applicable.

**Informed Consent Statement:** Not applicable.

**Data Availability Statement:** Data is contained within the article.

**Conflicts of Interest:** The authors declare no conflict of interest.

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
