# Peer review of "Fuzzy Quality Certification of Wheat"

_agriculture, doi:10.3390/agriculture12101640_

Round 1

Reviewer 1 Report

The authors present a fuzzy quality certification of wheat.

1. The author mainly introduces the experimental process of the proposed method. Are there other ways to evaluate wheat quality? If so, please add some latest algorithms for comparison.

2. The author writes in good English. Please carefully check the grammar and format of the full text to ensure that it meets the magazine's requirements.

3. Please use the same style for references.

Author Response

  1. The author mainly introduces the experimental process of the proposed method. Are there other ways to evaluate wheat quality? If so, please add some latest algorithms for comparison.

We completed the methodology with comparative approaches regarding wheat quality.

  1. The author writes in good English. Please carefully check the grammar and format of the full text to ensure that it meets the magazine's requirements.

We checked the grammar and the requirements of the magazine.

  1. Please use the same style for references.

Checked. We used Mendey software to insert references.

Reviewer 2 Report

The manuscript is very interesting. The following major improvements should be performed before acceptance.

1. Add your proposed work contributions clearly in the abstract.

2. Literature review is week. The authors didn't properly explore the literature. For instance;10.1007/s10462-021-10119-8; 10.3390/e23091176; 10.3390/math9172163.

3. I found some grammatical mistakes and typos in the manuscript. Please correct them. For example, On page 1, line 17, please correct the typos "fuzzzy". On page 2, lines 86-87, Please correct grammar mistakes.

4. Comparison with existing models is missing in the literature. Please add a comparison analysis between your launched study and some latest existing approaches in the literature.

5. Revise conclusion section. Also, add limitations of your work and some future directions.

6. How Matlab application is executed? Add Algorithm of Matlab and explain its time complexity.

7. You should use bullet symbol instead of negation on page 2, lines 48-51; and similarly all other places in the manuscript.

8. I think references 46 and 47 are same. Please correct this issue.

Author Response

  1. Add your proposed work contributions clearly in the abstract.

            I modify the abstract, presenting a more clearly the work contribution.

  1. Literature review is week. The authors didn't properly explore the literature. For instance;10.1007/s10462-021-10119-8; 10.3390/e23091176; 10.3390/math9172163.

I completed the literature review with the suggested sources.

  1. I found some grammatical mistakes and typos in the manuscript. Please correct them. For example, On page 1, line 17, please correct the typos "fuzzzy". On page 2, lines 86-87, Please correct grammar mistakes.

  1. Comparison with existing models is missing in the literature. Please add a comparison analysis between your launched study and some latest existing approaches in the literature.

I complete the article result section with comparison analysis between my proposed quality model and literature.

  1. Revise conclusion section. Also, add limitations of your work and some future directions.

I made the recommended additions.

  1. How Matlab application is executed? Add Algorithm of Matlab and explain its time complexity.

I completed the results section with references to the complexity of the algorithm.

  1. You should use a bullet symbol instead of negation on page 2, lines 48-51; and similarly all other places in the manuscript.

I made the recommended corrections.

  1. I think references 46 and 47 are the same. Please correct this issue.

I make the correction.

Reviewer 3 Report

Some significant findings should be discussed in the abstract.

Neutrosophic statistics is the extension of the fuzzy approach. The authors should explain the differences between them using some latest references.

The objective and novelty are not clear.

Need to add references to each equation 

Need to assign equation numbers to all equations.

The comparison section is missing

Limitations should be given

Assumptions of the study should be given

Neutrosophic statistics is the extension of classical statistics and is applied when the data is coming from a complex process or from an uncertain environment. The current study can be extended using neutrosophic statistics for future research. The statement that the proposed study can be extended to neutrosophic statistics can be added by citing some papers on neutrosophic statistics. 

Author Response

  1. Some significant findings should be discussed in the abstract.

I modify the abstract, presenting a more clearly the work contribution.

  1. Neutrosophic statistics is the extension of the fuzzy approach. The authors should explain the differences between them using some latest references.

I introduced references to the concept of neutrosophic statistics in the result section.

  1. The objective and novelty are not clear.

               I completed the obtained results with the description of the objectives and the novelty

  1. Need to add references to each equation

            I insert a reference for each equation.

  1. Need to assign equation numbers to all equations.

I insert a number for each equation.

  1. The comparison section is missing

            I complete the article result section with a comparison analysis between my proposed quality model and literature.

  1. Limitations should be given

I insert limitations and future direction in the conclusion section.

  1. Assumptions of the study should be given

I insert assumptions of the study in the result section.

Round 2

Reviewer 2 Report

The authors have revised properly according to my comments. So, I accept this version for publication.

Reviewer 3 Report

The authors have updated the paper but missed very close reference

Comparative Analysis of Climate Variability and Wheat Crop under Neutrosophic Environment,  MAPAN-Journal Metrology Society of India, 37(1):25–32